# Density Functional Theory Study of Methanol Steam Reforming on Pt_3_Sn(111) and the Promotion Effect of a Surface Hydroxy Group

**DOI:** 10.3390/nano14030318

**Published:** 2024-02-05

**Authors:** Ping He, Houyu Zhu, Qianyao Sun, Ming Li, Dongyuan Liu, Rui Li, Xiaoqing Lu, Wen Zhao, Yuhua Chi, Hao Ren, Wenyue Guo

**Affiliations:** 1College of Science, China University of Petroleum (East China), Qingdao 266580, China; s21090045@s.upc.edu.cn; 2School of Materials Science and Engineering, China University of Petroleum (East China), Qingdao 266580, China; b21140013@s.upc.edu.cn (D.L.); s21140007@s.upc.edu.cn (R.L.); luxq@upc.edu.cn (X.L.); zhaowen@upc.edu.cn (W.Z.); chiyuhua@upc.edu.cn (Y.C.); renh@upc.edu.cn (H.R.); 3SINOPEC Dalian Research Institute of Petroleum and Petrochemicals Co., Ltd., Dalian 116045, China; sunqianyao.fshy@sinopec.com (Q.S.); liming.fshy@sinopec.com (M.L.)

**Keywords:** methanol, steam reforming, PtSn alloy, hydroxy group, density functional theory

## Abstract

Methanol steam reforming (MSR) is studied on a Pt_3_Sn surface using the density functional theory (DFT). An MSR network is mapped out, including several reaction pathways. The main pathway proposed is CH_3_OH + OH → CH_3_O → CH_2_O → CH_2_O + OH → CH_2_OOH → CHOOH → COOH → COOH + OH → CO_2_ + H_2_O. The adsorption strengths of CH_3_OH, CH_2_O, CHOOH, H_2_O and CO_2_ are relatively weak, while other intermediates are strongly adsorbed on Pt_3_Sn(111). H_2_O decomposition to OH is the rate-determining step on Pt_3_Sn(111). The promotion effect of the OH group is remarkable on the conversions of CH_3_OH, CH_2_O and *trans*-COOH. In particular, the activation barriers of the O–H bond cleavage (e.g., CH_3_OH → CH_3_O and *trans*-COOH → CO_2_) decrease substantially by ~1 eV because of the involvement of OH. Compared with the case of MSR on Pt(111), the generation of OH from H_2_O decomposition is more competitive on Pt_3_Sn(111), and the presence of abundant OH facilitates the combination of CO with OH to generate COOH, which accounts for the improved CO tolerance of the PtSn alloy over pure Pt.

## 1. Introduction

Methanol steam reforming (MSR) has been widely accepted as a candidate method of generating hydrogen for the on-board application of direct methanol fuel cells (DMFCs) [1,2]. Platinum (Pt) is generally applied as a DMFC catalyst because of its thermal stability and high catalytic activity [2,3,4,5,6]. However, CO molecules are primarily produced from methanol (CH_3_OH) decomposition and gradually accumulate on Pt, which ultimately leads to CO poisoning and the loss of activity of Pt catalysts [7,8]. Alloying is an effective way to enhance the resistance of metal catalysts. Recently, a PtSn alloy showed promise as an efficient DMFC catalyst with considerable CH_3_OH electrocatalytic rates compared to Pt [8,9,10,11,12,13,14,15], and it is reported to be active for CO oxidation [16,17,18]. Therefore, an in-depth study of MSR reactions on PtSn is an essential prerequisite to rationally design more efficient and stable PtSn-based catalysts for DMFC applications.

Generally, the MSR process can be summarized as the following two main reaction mechanisms based on previous experimental research studies [19,20,21]. The first mechanism (M1) proceeds with the direct dehydrogenation of CH_3_OH and the formation of CO; then, CO is oxidized to CO_2_ via the water–gas shift (WGS) reaction (H_2_O + CO → H_2_ + CO_2_) [22,23]. The second mechanism (M2) includes the reactions of intermediates with adsorbed OH, which is generated from water decomposition (H_2_O → OH + H), to yield CH_2_OO, CHOOH, CHOO and, finally, H_2_ and CO_2_ [24,25,26]. From the perspective of theoretical research, different catalyst models account for different intermediates and MSR mechanisms. Using density functional theory (DFT) calculations, Luo et al. [27] investigated MSR reactions on Co(0001) and Co(111), and their results showed that the direct decomposition of CH_2_O to CO is favored rather than CH_2_OOH formation, indicating the preference of the M1 mechanism. Fajín and Cordeiro [28] performed a DFT investigation on bimetallic Ni−Cu alloy surfaces and also confirmed the M1 mechanism. They found that the MSR evolves mostly through CH_3_OH decomposition followed by the WGS reaction. In these studies, the surface OH group did not take part in the main reaction pathway, but it can become involved in or influence the MSR process on other metal and alloy surfaces. Lin et al. proposed that MSR reactions on Cu(111) [25,29] and PdZn(111) [26,30] follow the M2 mechanism, that is, the stepwise dehydrogenation of CH_3_OH occurs first, followed by CH_2_O formation; then, CH_2_O combines with OH, which produces a CH_2_OOH intermediate. Finally, CH_2_OOH is further dehydrogenated to yield CO_2_. CH_3_O dehydrogenation is identified as the rate-determining step on both Cu(111) and PdZn(111) surfaces. Li et al. [31] also confirmed the M2 mechanism of the MSR on an α-MoC(100) surface using DFT calculations. The results suggest that the stepwise O−H and C−H bond scissions of CH_3_OH yield CH_2_O. Then, CH_2_OOH is formed through the combination of CH_2_O and OH, which is preferred over the decomposition path of CH_2_O to CHO and H. In addition to its direct involvement in the MSR reaction pathway, the surface OH group can also exert an important influence on the MSR process. Huang et al. [32] studied CH_3_OH decomposition on PdZn(111) using the DFT and found that the presence of co-adsorbed OH species would hinder C–H bond scission while significantly reducing the energy barrier of the O–H bond scission. Thus, CH_3_OH preferentially undergoes O–H bond scission to form CH_3_O because of the influence of OH. Although a great number of efforts have been made to determine the MSR mechanisms of various catalyst models, the detailed MSR process, as well as intermediate information, has not been unambiguously elucidated for specific new catalyst models. At present, there are no theoretical reports available to elucidate the complete MSR mechanism on a PtSn alloy surface. Furthermore, the effect of OH species on the MSR process should also be clarified.

In this work, a periodic DFT investigation is carried out to elucidate the MSR mechanism on a PtSn alloy surface. Among Pt_x_Sn catalysts with different Sn contents, Pt_3_Sn has been proven to have the best performance for the oxidation of methanol and CO in DMFCs [17,18]. Thus, Pt_3_Sn(111) is chosen as a representative PtSn alloy for DFT calculations. The adsorption structures, elementary reactions and potential energy surfaces (PESs) are illustrated for methanol decomposition and steam reformation processes, and the effect of the OH group on the catalytic mechanism is discussed in detailed.

## 2. Computational Methods

DFT calculations were conducted using the DMol^3^ program package [33,34,35]. Exchange and correlation effects were treated using the GGA-PW91 functional [36,37,38]. The DSPP method [39] was applied for Pt and Sn atoms, while C, H and O atoms were treated with an all-electron basis set. The valence electron functions were expanded into a set of numerical atomic orbitals on a double-numerical basis with polarization functions. A Fermi smearing of 0.005 Hartree and a real-space cutoff of 4.5 Å were used. Spin-polarization was applied in all calculations.

The lattice constant of the Pt_3_Sn was calculated to be 4.01 Å, in good agreement with the experimental value of 4.00 Å [40]. The Pt_3_Sn(111) surface was built using a p(2 × 2) unit cell with a four-layer slab, and each layer consisted of three Pt atoms and one Sn atom. The height of the vacuum region was set at 12 Å. The reciprocal space was sampled with a (5 × 5 × 1) *k*-points grid generated automatically using the Monkhorst–Pack method [41]. The uppermost two layers of the slab were relaxed with adsorbates, while two substrate layers were fixed at bulk positions. 

High-symmetry sites on Pt_3_Sn(111) are presented in Figure 1. The adsorption energies (*E*_ads_) were calculated as follows: [42,43]
*E*_ads_ = *E*_adsorbate_ + *E*_slab_ − *E*_adsorbate/slab_(1)
where *E*_adsorbate/slab_ is the energy of the adsorbate/slab adsorption system, and *E*_adsorbate_ and *E*_slab_ are the energies of the free adsorbate and the clean slab, respectively. By this definition, stable adsorption will have a positive adsorption energy. 

Transition state (TS) searches were performed at the same theoretical level using the complete linear synchronous transit/quadratic synchronous transit (LST/QST) method [22,23,24,44]. In this method, an LST maximization was performed, followed by an energy minimization in directions conjugating to the reaction pathway to obtain an approximated TS. The approximated TS was used to perform a QST maximization, and then another conjugated gradient minimization was performed. This cycle was repeated until a stationary point was located. The convergence criterion for the TS searches was set to 0.01 hartree/Å for the root mean square of the atomic forces. The energy barrier (*E*_a_) was determined as the energy difference between the corresponding TS and the initial state (IS), and the reaction energy (*E*_r_) was defined as the energy difference between the final state (FS) and the IS.

## 3. Results and Discussion

### 3.1. Adsorption Structures and Energies

Figure 2 shows the most stable adsorption geometries of intermediates in MSR, and Table 1 shows the corresponding adsorption energies (*E*_ads_) and geometric parameters. For clarity, the geometries and energies of the sub-stable adsorptions of the involved intermediates are presented in Appendix A. In our previous study of methanol decomposition on Pt_3_Sn(111) [43], several intermediates were calculated in detail, and the most stable adsorption sites along with *E*_ads_ can be summarized as follows CH_3_OH at T^Sn^ (0.47 eV), CH_3_O at T^Sn^ (1.71 eV), CH_2_OH at T^Pt^ (1.94 eV), CH_2_O at F^2PtSn^ (0.38 eV), CHOH at B^2Pt^ (3.14 eV), CHO at T^Pt^ (2.28 eV), COH at H^3Pt^ (4.05 eV), CO_2_ at B^PtSn^ (0.11 eV), OH at B^2Pt^ (2.51 eV) and O at F^2PtSn^ (4.12 eV). In this work, we focus on the reformation process, especially the OH-involved paths. Accordingly, the reaction intermediates of MSR are described in detail below.

Carboxymethyl (CH_2_OOH) is formed through the combination of CH_2_O and an OH group and preferentially adsorbs at a bridge site via the *η*^1^(O)-*η*^1^(O) mode, which is different from the unidentate *η*^3^(O) modes at the hollow sites of Cu(111) [25], PdZn(111) [26] and Co(111) [27]. The *E*_ads_ values of CH_2_OOH are 1.89 eV (B^PtSn^) and 1.65 (B^2Pt^), respectively. At the B^PtSn^ site (Figure 2), two C–O bond lengths are 1.35 and 1.52 Å, and the O–Sn and O–Pt distances are 2.15 and 2.31 Å, respectively. Dioxomethylene (CH_2_OO) was reported to adsorb at a bridge site in a bidentate *η*^1^(O)-*η*^1^(O) mode on Cu(111), PdZn(111) and Co(111) surfaces [25,26,27]. However, we found that CH_2_OO has two adsorption modes on Pt_3_Sn(111) which are the *η*^2^(O)-*η*^1^(O) mode at the F^2PtSn^ site and the *η*^1^(O)-*η*^1^(O) mode at the B^PtSn^ site. As listed in Table 1, the *η*^2^(O)-*η*^1^(O) mode (Figure 2) is more stable with an *E*_ads_ of 3.24 eV, and the two O–Pt and O–Sn distances are 2.09, 2.26 and 2.27 Å, respectively. The *E*_ads_ of the *η*^1^(O)-*η*^1^(O) mode (Appendix A) was calculated to be 3.08 eV, consistent with the previous DFT result for CH_2_OO adsorption via the same *η*^1^(O)-*η*^1^(O) mode on Cu(111) [25]. For Formic acid (CHOOH), the most stable adsorption site is T^Sn^, and the corresponding *E*_ads_ is 0.49 eV. The molecule plane of CHOOH is almost vertical with the OH group pointing down toward the surface (Figure 2). The two C–O bond lengths are 1.23 and 1.32 Å, respectively. The CHOOH at the T^Pt^ has a similar adsorption configuration (Appendix A) with a lower *E*_ads_ of 0.38 eV. The other four adsorption geometries of CHOOH at F^2PtSn^, F^3Pt^, H^2PtSn^ and H^3Pt^ involve molecule planes almost parallel to Pt_3_Sn(111) with ~3.70 Å above the surface (Appendix A). Formate (CHOO) can adsorb at B^2Pt^ and B^PtSn^ with the *η*^1^(O)-*η*^1^(O) mode, and the B^PtSn^ site is preferred. At the B^PtSn^ site, the molecular plane is perpendicular to the Pt_3_Sn(111), with an *E*_ads_ of 2.52 eV; the O–Pt and O–Sn distances are 2.17 and 2.29 Å, respectively (Figure 2). At the B^2Pt^ site, the *E*_ads_ decreases to 2.08 eV. Carboxyl (COOH) has two isomers which are *cis-* and *trans*-COOH, respectively [25]. The *cis*-COOH can adsorb at the T^Pt^, T^Sn^ and T^2Pt^ sites with corresponding *E*_ads_ values of 2.48, 1.26 and 2.37 eV, respectively. The T^Pt^ site can thus be identified as the most stable binding site for *cis*-COOH; the molecular plane is nearly perpendicular to Pt_3_Sn(111), with C–Pt and two C–O bond lengths of 2.03, 1.22 and 1.37 Å, respectively (Figure 2). For *trans*-COOH, the *E*_ads_ values are 2.35 (T^Pt^), 1.09 (T^Sn^), 2.39 (T^2Pt^) and 2.41 (T^PtSn^) eV. The *cis* isomer binds slightly more strongly to Pt_3_Sn(111) than its *trans* counterpart (2.48 vs. 2.41 eV), similar to COOH adsorption on Cu(111) [25]. CO_2_ adsorbs weakly above the B^PtSn^ and B^2Pt^ sites with the same *E*_ads_ value of 0.11 eV. At bridge sites, this linear molecule lies almost parallel to the Pt_3_Sn(111) at a distance of ~4.00 Å above the surface (Figure 2). These results are consistent with those of previous DFT studies of weak CO_2_ adsorptions over Cu(111) [25], Co(0001) [45] and Co(111) [27]. H_2_O adsorbs above the T^Sn^ site via the O–Sn bond, and the two O–H axes are parallel to the Pt_3_Sn(111) surface (Figure 2) with bond lengths of 0.98 Å. The binding strength of H_2_O is very weak, mirrored by a low *E*_ads_ of 0.01 eV, which is also consistent with weak H_2_O adsorption on Cu(111) [25], Co(111) [27] and Co(0001) [45]. The most stable sites and *E*_ads_ values for intermediates via *η*(O) can be summarized as followed: CH_2_OOH at B^PtSn^ (1.89 eV), CH_2_OO at F^2PtSn^ (3.24 eV), HCOOH at T^Sn^ (0.49 eV), CHOO at B^PtSn^ (2.52 eV), *cis*-COOH at T^Pt^ (2.48 eV), *trans*-COOH at B^PtSn^ (2.41 eV) and H_2_O at T^Sn^ (0.01 eV). Taking into account the adsorption properties of other intermediates (CH_3_OH, CH_2_OH, CH_3_O, CHOH, CH_2_O, COH, CHO, etc.) [43], Sn strengthens the binding of these intermediates to the Pt_3_Sn(111) surface via *η*(O).

### 3.2. Elementary Reaction Steps

The decomposition reactions of CH_3_OH, CH_2_OH, CH_3_O, CH_2_O and CHO via O−H, C−H and C−O bond scissions were calculated in our previous study [43]. We found that CH_3_OH decomposition began with O−H bond scission, followed by C−H bond cleavages, that is, CH_3_OH → CH_3_O → CH_2_O → CHO → CO. To identify the optimal MSR pathway, multiple reactions were further investigated in this work, including H_2_O dissociation into OH and H and subsequent OH-involving reactions with CH_3_OH and its dehydrogenated intermediates. The configurations of the involved IS, TS and FS are presented in Figure 3 and Figure 4. Sixteen reactions (R1–R16) were considered in total with their thermodynamic and kinetic parameters.

H_2_O Activation. In the IS, H_2_O adsorbs weakly above the T^Sn^ site. For the reaction R1, the O–H bond is ruptured, with the H atom migrating toward the adjacent Pt atom. The O–H distance of H_2_O is elongated from 0.98 Å in the IS to 1.62 Å in TS1, as shown in Figure 3. Finally, the OH binds at the B^2Pt^ site, and the H sits at the H^3Pt^ site. This reaction is exothermic by 0.47 eV, with an energy barrier of 0.97 eV. For comparison, the *E*_a_ of H_2_O decomposition on Pt_3_Sn is much lower than that on Cu(111) (1.11 eV) [25]. 

CH_3_OH + OH. In reaction R2, CH_3_OH and OH adsorb at the T^Sn^ and T^Pt^ sites in the IS, respectively, and in the FS, CH_3_O and H_2_O locate at the same sites as in the IS. In TS2 (Figure 3), the distance of the breaking O−H bond in CH_3_OH is 1.24 Å, smaller than that in the direct dehydrogenation of CH_3_OH (0.97 Å) [43] This step is slightly exothermic by 0.04 eV, and the *E*_a_ is only 0.02 eV, which is 0.97 eV lower than direct methanol dehydrogenation by O−H bond cleavage at the same site of the T^Sn^ [43]. 

CH_x_O + OH (x = 0–3). In reaction R3 of CH_3_O with OH, the energy barrier is 0.84 eV with a reaction energy of −0.87 eV. In TS3, the distance of the breaking O−H bond is 1.27 Å. In reaction R4, the CH_2_O fragment is weakly bound at the F^2PtSn^, while the OH fragment stays at T^Pt^ site, yielding CH_2_OOH at the B^2Pt^ site. In TS4, two fragments move to the T^Pt^ site, and the distance of the cleaved O−H bond is 2.09 Å. This step is exothermic by 0.45 eV and has an activation barrier of 0.43 eV, lower than that of 0.75 eV for CH_2_O → CHO [25]. A similar process also occurs on Cu(111) [25] and PdZn(111) [26] For reaction R5, co-adsorbed CHO at the H^3Pt^ site and OH at the T^Sn^ site are taken as the IS, and the HCOOH at the H^3Pt^ site is the FS. The distance between C and O atoms is shortened from 3.67 Å in the IS to 1.92 Å in TS5 and to 1.36 Å in the FS. This reaction has an activation barrier of 0.63 eV with an exothermicity of 0.70 eV. For reaction R6, the IS is the co-adsorption of OH at the T^Sn^ site and CO at the T^Pt^ site, and the FS is COOH at the T^Pt^ site. In TS6, the distance of the forming C−O bond is 1.91 Å. This step is exothermic by 0.25 eV, with an activation barrier of 0.39 eV. 

CH_2_OOH dehydrogenation. Two reaction pathways exist for CH_2_OOH dehydrogenation. The first is C–H bond scission (R7, CH_2_OOH → CHOOH + H), producing a CHOOH fragment above the B^PtSn^ site with H at the H^3Pt^ site. For TS7 (Figure 3), the C−H distance of the breaking C−H bond is 1.53 Å, which stretches from 1.11 Å in the IS to 3.95 Å in the FS. This reaction has an activation barrier of 0.40 eV with a reaction energy of −0.74 eV. The second is O–H bond scission (R8, CH_2_OOH → CH_2_OO + H), which starts with the CH_2_OOH at the B^2Pt^ site and ends with a co-adsorbed CH_2_OO fragment at the B^2Pt^ site and H at the T^Pt^ site. In TS8 (Figure 3), the O–H distance of the breaking O–H bond is 1.49 Å. This reaction has a higher activation barrier of 1.64 eV and is endothermic by 0.91 eV. Based on thermodynamic and kinetic viewpoints, CH_2_OOH dehydrogenation on Pt_3_Sn(111) tends to yield CHOOH rather than CH_2_OO, that is, the C–H bond scission of reaction R7 is more competitive than the O–H bond scission of reaction R8.

CH_2_OO and CHOOH dehydrogenation. CH_2_OO dehydrogenation, denoted as reaction R9, yields bidentate CHOO binding at the B^PtSn^ site and H at the T^Pt^ site (Figure 4). This step has a low activation barrier of 0.37 eV and a high exothermicity of 1.55 eV. For TS9, H moves down and locates above the B^2Pt^ site, while CHOO remains at the B^PtSn^ site; the C–H distance of the breaking C–H bond is 1.11 Å. CHOOH dehydrogenation includes C–H bond scission (reaction R10) and O–H bond cleavage (reaction R11). The C–H bond cleavage of CHOOH yields a B^PtSn^-site-adsorbed COOH fragment and a T^Pt^-site-adsorbed H atom (Figure 4). This reaction involves an energy barrier of 0.44 eV and an endothermicity of 0.01 eV. For TS10, the breaking C–H bond is elongated to 2.07 Å. The O–H bond cleavage of CHOOH is slightly exothermic by 0.02 eV, and the activation barrier is 0.78 eV. For TS11, the O–H bond is elongated by 1.35 Å, and the leaving H adsorbs at the B^2Pt^ site. In the FS, CHOO binds to the Pt_3_Sn(111) surface in a bidentate configuration, and the detached H locates at the T^Pt^ site.

CHOO and COOH dehydrogenation. The CHOO is produced from CH_2_OO dehydrogenation or CHOOH dehydrogenation via the O–H bond cleavage. The further dehydrogenation of CHOO generates CO_2_ and an H atom, which is denoted as reaction R12 (Figure 4). In TS12, the C–H distance of the breaking C–H bond is 2.40 Å. After the C–H bond scission, the detached H atom adsorbs at the T^Pt^ site, while the CO_2_ adsorbs above the B^PtSn^ site. This reaction is exothermic by 0.31 eV, and the activation barrier is 1.06 eV. The dehydrogenation process of CHOO could also be accomplished with assistance from an adsorbed OH group (reaction R13). This step starts with co-adsorbed CHOO at the B^PtSn^ site and OH at the T^Pt^ site and ends with weakly bonded CO_2_ and H_2_O on the surface. The activation barrier of this step is 1.53 eV, and the reaction energy is −1.10 eV. Compared with the direct dehydrogenation of CHOO (R12), the OH-assisted reaction of CHOO with OH to H_2_O and CO_2_ (R13) has a relatively higher energy barrier, suggesting that R12 is more favorable than R13. The isomerization of *cis*-COOH to form *trans*-COOH (R14) is necessary for COOH dehydrogenation because the O–H bond of the adsorbed COOH points away from the surface in the *cis*-mode but swings toward the surface in the *trans*-mode, which is helpful for O–H bond activation. This isomerization step involves an energy barrier of 0.53 eV. Subsequently, CO_2_ is produced by removing the H atom from *trans*-COOH (R15), which accounts for an activation barrier of 1.04 eV and a reaction energy of −0.23 eV. For TS15, the O–H distance of the breaking O–H bond is 1.38 Å, and the CO_2_ is above the B^PtSn^ site and an H atom locates at the T^Pt^ site. Similar to CHOO, *trans*-COOH can also react with OH to generate H_2_O and CO_2_ (R16). This step starts with co-adsorbed *trans*-COOH at the T^Pt^ site and OH at the B^PtSn^ site and ends with CO_2_ above the B^PtSn^ site and H_2_O above the T^Pt^ site. This OH-assisted step is exothermic by 0.74 eV, with a lower activation barrier of 0.11 eV.

### 3.3. MSR Mechanisms

Based on the calculated results, the potential energy surfaces of MSR on Pt_3_Sn(111) are presented in Figure 5. CH_3_OH decomposition with the assistance of OH to form CH_3_O + H_2_O and CH_3_OH dehydrogenation via O–H bond scission to form CH_3_O + H involve activation barriers of 0.02 and 1.01 eV, respectively. Compared with the direct dehydrogenation of CH_3_OH to CH_3_O on Pt_3_Sn(111), the involvement of the OH group greatly promotes this dehydrogenation step. For the intermediate CH_3_O, however, the OH group is not helpful for C–H bond cleavage because the direct dehydrogenation of CH_3_O to CH_2_O only needs to overcome an activation barrier of 0.42 eV compared with the case of CH_3_O + OH → CH_2_O + H_2_O (*E*_a_ = 0.84 eV). For the intermediate CH_2_O, the transition state of the C–H bond activation with the participation of the OH group was not found in spite of an elaborate search. CH_2_O has two competitive paths: the direct dehydrogenation, CH_2_O → CHO (*E*_a_ = 0.75 eV), and a combination with the OH group, CH_2_O + OH → CH_2_OOH (*E*_a_ = 0.43 eV). Therefore, the combination of CH_2_O with OH is more favorable. The further dehydrogenation of the newly formed CH_2_OOH has two possibilities, which are O–H and C–H bond activations. We found that the C–H bond scission of CH_2_OOH → CHOOH + H (*E*_a_ = 0.40 eV) is more competitive than the O–H bond cleavage of CH_2_OOH → CH_2_OO + H (*E*_a_ = 1.64 eV). Similar to CH_2_OOH, the intermediate CHOOH also tends to break the C–H bond (*E*_a_ = 0.44 eV) rather than the O–H bond (*E*_a_ = 0.78 eV). CHOOH dehydrogenation yields *cis*-COOH, followed by an isomerization step toward *trans*-COOH. Compared with *cis*-COOH, the adsorption geometry of *trans*-COOH is favored for O–H bond activation: *trans*-COOH → CO_2_ + H (*E*_a_ = 1.04 eV). The participation of the OH group substantially reduces the dehydrogenation barrier of *trans*-COOH via *trans*-COOH + OH → CO_2_ + H_2_O (*E*_a_ = 0.11 eV), indicating the promotion effect of the OH group. 

Figure 6 summarizes the MSR reaction network based on the direct decomposition of methanol in our previous work [43] and the results calculated in this study. The most favorable pathway follows the M2 mechanism, in which important intermediates were identified as follows: CH_3_OH + OH → CH_3_O + H_2_O (2)
CH_3_O → CH_2_O + H(3)
CH_2_O + OH → CH_2_OOH(4)
CH_2_OOH → CHOOH + H(5)
CHOOH → COOH + H(6)
COOH + OH → CO_2_ + H_2_O(7)
H_2_ production originates from H_2_O decomposition and the dehydrogenation of important intermediates (CH_3_O, CH_2_OOH and CHOOH). The promotion effect of the surface OH group on the conversions of CH_3_OH, CH_2_O and *trans*-COOH is remarkable. In particular, the energy barriers of the O–H bond activation (e.g., CH_3_OH → CH_3_O and *trans*-COOH → CO_2_) decrease substantially by ~1 eV due to the involvement of the surface OH group, while OH fails to facilitate C–H bond activation. The above results are consistent with previous DFT calculations of CH_3_OH decomposition by Huang et al. [32] in which the presence of a surface OH group on PdZn(111) impeded the C–H bond scission of CH_3_OH but substantially decreased the O–H bond-activation barrier. For comparison, Jin et al. [46] found that the OH group on Pt(111) could also be beneficial to MSR reactions, such as CH_3_OH → CH_3_O and CH_2_O + OH → CH_2_OOH. However, it is relatively difficult to dissociate water and generate the OH group on Pt(111) compared with the direct dehydrogenation of CH_3_OH. The OH group is only available when the difference in the energy barrier between H_2_O decomposition and CH_3_OH dehydrogenation is comparable. Thus, the MSR process on Pt(111) still follows the M1 mechanism, which is stepwise CH_3_OH decomposition to CO followed by WGS reactions: CH_3_OH → CH_2_OH → CHOH → CHO → CO → CO + OH → COOH. In this study, the Pt_3_Sn(111) surface reduced the difference in the *E*_a_ between H_2_O → H + OH (*E*_a_ = 0.97 eV) and CH_3_OH → CH_3_O + H (*E*_a_ = 1.01 eV)/CH_3_OH → CH_2_OH + H (*E*_a_ = 1.09 eV). The relatively lower *E*_a_ of H_2_O decomposition indicates the availability of the OH group, which facilitates the MSR process. The initial H_2_O decomposition to the OH group involves the highest activation barrier of 0.97 eV through the main reaction pathway. Thus, H_2_O decomposition could be identified as the rate-determining step for MSR on Pt_3_Sn(111) rather than the commonly accepted C–H bond-cleavage steps such as CH_3_OH → CH_2_OH on Pt(111) [46] and CH_3_O → CH_2_O on both Cu(111) [25] and PdZn(111) [26]. Compared with the dehydrogenation reactions of CH_3_OH, the initial H_2_O → OH step involves relatively higher selectivity on Pt_3_Sn, which accounts for the improved CO tolerance of PtSn alloys over pure Pt. 

## 4. Conclusions

DFT calculations were performed to investigate possible intermediates and MSR reaction pathways on Pt_3_Sn(111). The MSR network was mapped out. The most favorable pathway was identified as follows: CH_3_OH + OH → CH_3_O → CH_2_O → CH_2_O + OH → CH_2_OOH → CHOOH → COOH → COOH + OH → CO_2_ + H_2_O. Along this main reaction pathway, the adsorption strengths of CH_3_OH, CH_2_O, CHOOH, H_2_O and CO_2_ are relatively weak (*E*_ads_ < 0.5 eV), while other intermediates are strongly adsorbed at the T^Sn^ site for CH_3_O (*E*_ads_ = 1.71 eV), at the T^Pt^ site for *cis*-COOH (*E*_ads_ = 2.48 eV) and at the B^PtSn^ site for CH_2_OOH (*E*_ads_ = 1.89 eV) and *trans*-COOH (*E*_ads_ = 2.41 eV). H_2_ production originates from H_2_O decomposition and the dehydrogenation of important intermediates (CH_3_O, CH_2_OOH and CHOOH). H_2_O decomposition into OH involves an activation barrier of 0.97 eV and was identified as the rate-determining step for the MSR process on Pt_3_Sn(111). The promotion effect of the surface OH group on the conversions of CH_3_OH, CH_2_O and *trans*-COOH is remarkable. In particular, the energy barriers of the O–H bond activation (e.g., CH_3_OH → CH_3_O and *trans*-COOH → CO_2_) decrease substantially by ~1 eV due to the involvement of the surface OH group. Compared with the case on Pt(111), the formation of a surface OH group from H_2_O decomposition is more competitive on Pt_3_Sn(111), and the presence of abundant OH facilitates the combination of CO with OH to generate COOH, which accounts for the improved CO tolerance of PtSn alloys over pure Pt.

## Figures and Tables

**Figure 1 nanomaterials-14-00318-f001:**
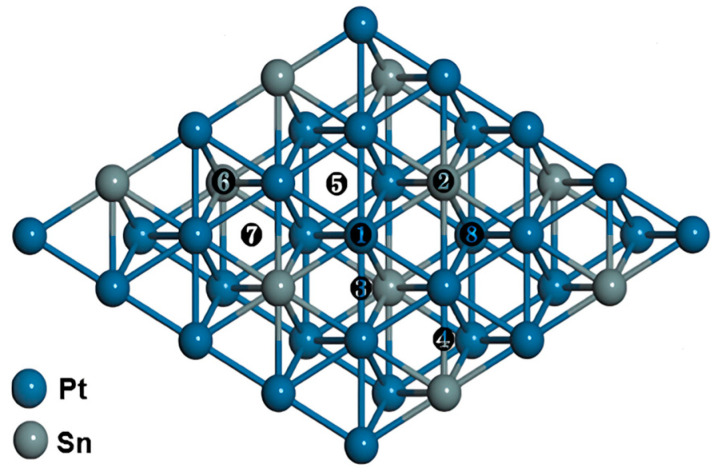
High-symmetry adsorption sites on Pt_3_Sn(111). The labels ①~⑧ represent the top of the Pt site (T^Pt^), the top of the Sn site (T^Sn^), the Pt-Pt bridge site (B^2Pt^), the Pt-Sn bridge site (B^PtSn^), the fcc/hcp site consisting of three surface nearest-neighboring Pt atoms (F^3Pt^/H^3Pt^) and the fcc/hcp site consisting of one Sn and two Pt atoms (F^2PtSn^/H^2PtSn^), respectively.

**Figure 2 nanomaterials-14-00318-f002:**
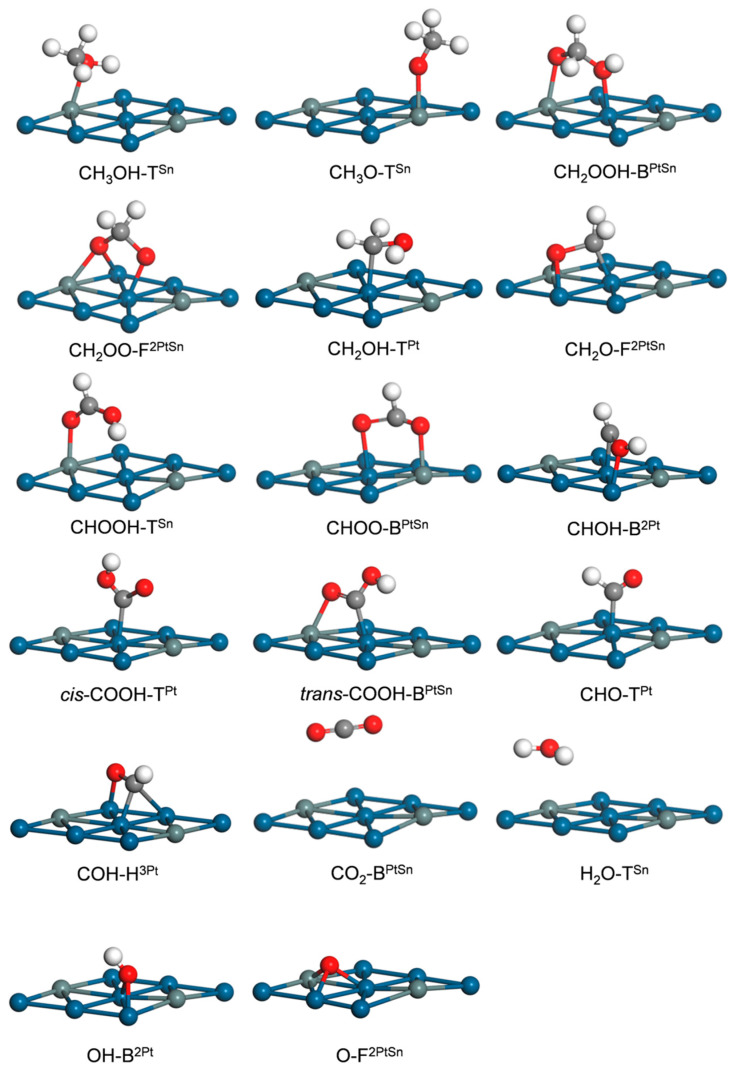
Stable adsorption structures of MSR intermediates on Pt_3_Sn(111). The C, H, O, Pt and Sn atoms are denoted as gray, white, red, blue and gray balls, respectively.

**Figure 3 nanomaterials-14-00318-f003:**
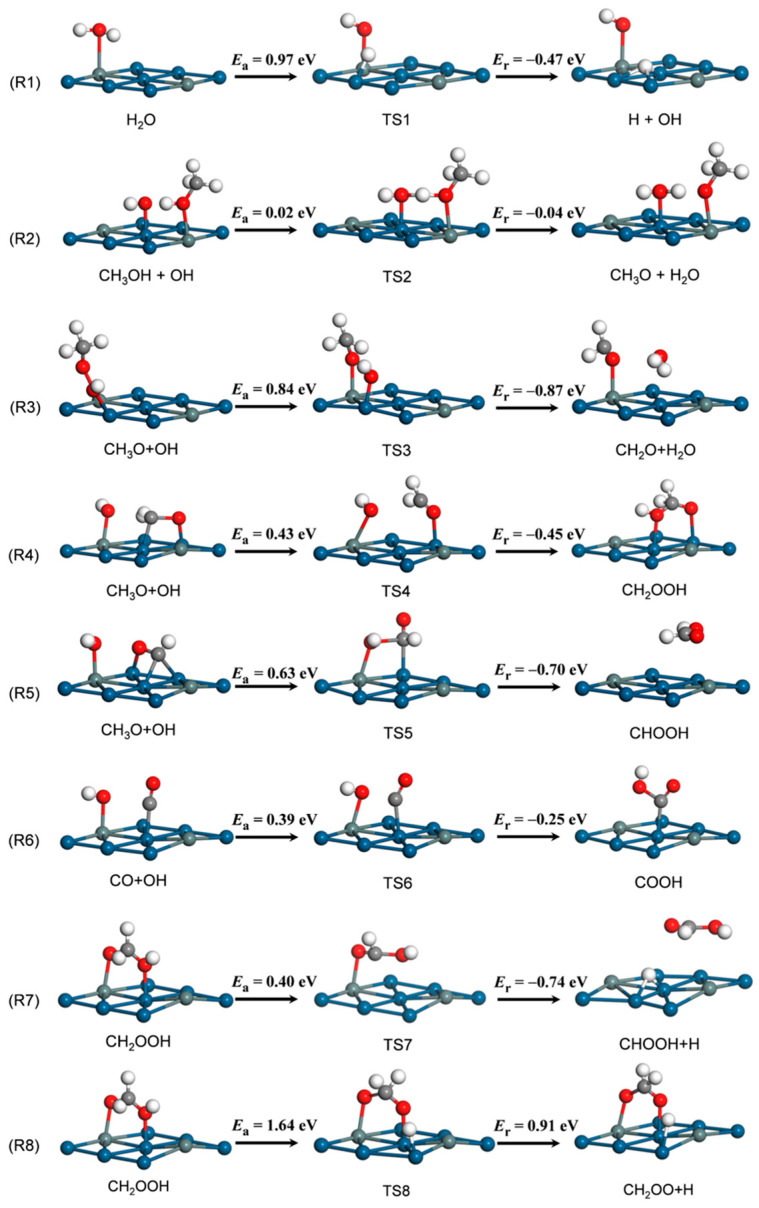
MSR reactions involving OH (R1–R8) on Pt_3_Sn(111). Parameters follow the same notation as in Figure 2.

**Figure 4 nanomaterials-14-00318-f004:**
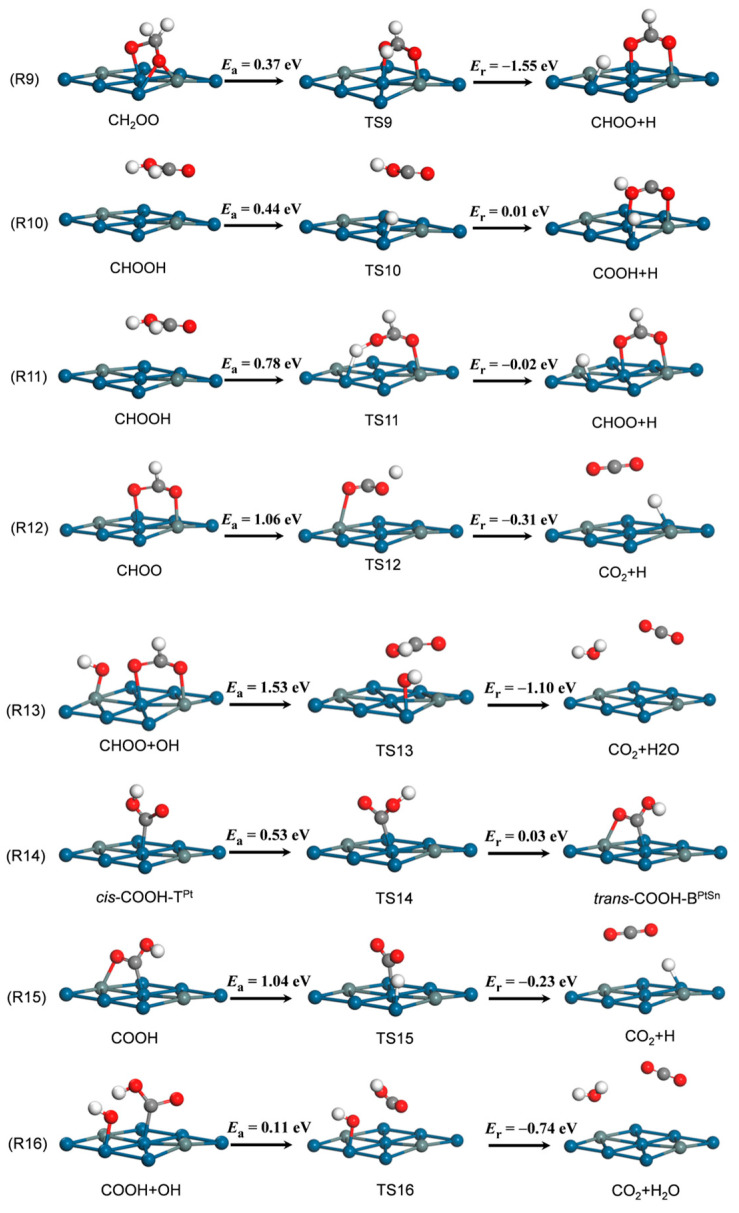
The MSR reactions involving OH (R9–R16) on Pt_3_Sn(111). Parameters follow the same notation as in Figure 2.

**Figure 5 nanomaterials-14-00318-f005:**
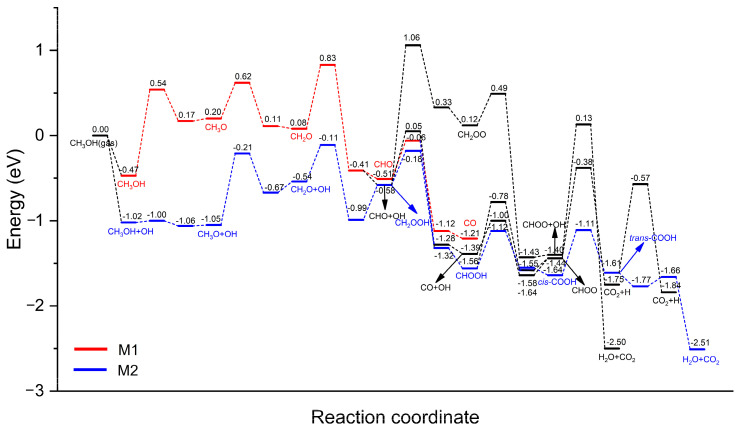
Potential energy surface (PES) of MSR on Pt_3_Sn(111). The detailed reaction pathways of the M1 and M2 mechanisms are shown in red and blue colors, respectively. Data on the direct decomposition of methanol (CH_3_OH → CH_3_O → CH_2_O → CHO → CO) were taken from our precious work [43].

**Figure 6 nanomaterials-14-00318-f006:**
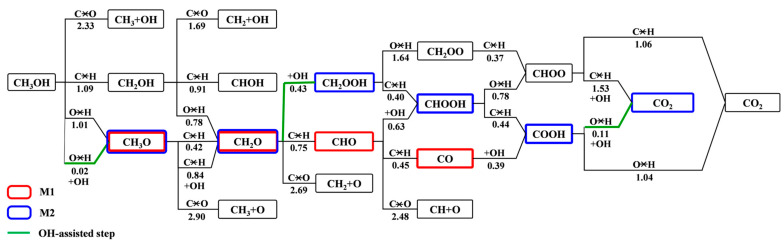
Proposed detailed MSR pathways on Pt_3_Sn(111). The M1 and M2 mechanisms are denoted by red and blue boxes, respectively. The OH-assisted steps are marked with green lines. The generated H and H_2_O are omitted for clarity.

**Table 1 nanomaterials-14-00318-t001:** The most stable adsorption sites, geometric parameters (in Å) and energies (in eV) for MSR intermediates on Pt_3_Sn(111).

Species	Site	Mode	*d* _C/O-Pt/Sn_	*E* _ads_
CH_3_OH	T^Sn^	*η*^1^(O)	2.64	0.47
CH_3_O	T^Sn^	*η*^1^(O)	2.05	1.71
CH_2_OOH	B^PtSn^	*η*^1^(O)-*η*^1^(O)	2.15, 2.31	1.89
CH_2_OO	F^2PtSn^	*η*^2^(O)-*η*^1^(O)	2.09, 2.26, 2.27	3.24
CH_2_OH	T^Pt^	*η*^1^(C)	2.14	1.94
CH_2_O	F^2PtSn^	*η*^1^(C)-*η*^2^(O)	2.13, 2.28, 2.43	0.38
HCOOH	T^Sn^	*η*^1^(O)	2.58	0.49
CHOO	B^PtSn^	*η*^1^(O)-*η*^1^(O)	2.17, 2.29	2.52
CHOH	B^2Pt^	*η*^2^(C)	2.09, 2.12	3.14
*cis*-COOH	T^Pt^	*η*^1^(C)	2.03	2.48
*trans*-COOH	B^PtSn^	*η*^1^(C)-*η*^1^(O)	2.04, 2.55	2.41
CHO	T^Pt^	*η*^1^(C)	2.01	2.28
COH	H^3Pt^	*η*^3^(C)	2.04, 2.06, 2.11	4.05
CO_2_	B^PtSn^	-	-	0.11
H_2_O	T^Sn^	-	-	0.01
OH	B^2Pt^	*η*^2^(O)	2.23, 2.24	2.51
O	F^2PtSn^	*η*^3^(O)	2.13, 2.14, 2.14	4.12

## Data Availability

Data are contained within the article.

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
