# Peer review of "Density Functional Theory Study of Methanol Steam Reforming on Pt3Sn(111) and the Promotion Effect of a Surface Hydroxy Group"

_nanomaterials, 2024, doi:10.3390/nano14030318_

Round 1
Reviewer 1 Report
Comments and Suggestions for Authors
As it is known, and also supported by the Authors “Methanol steam reforming (MSR) has emerged as a leading candidate to generate hydrogen fuel……” Unfortunately, it is not reflected in the manuscript under evaluation. According to the chain of reactions proposed by the Authors based on performed calculations, the products of the MSR process are …… water and carbon dioxide. ???????????????? What about hydrogen? Authors should indicate if and where hydrogen is generated.
Figure 1 is completely illegible and fuzzy. The resolution of Figure 1 should be increased. Are only platinum and tin atoms present in this drawing? In general, all the figures are of very low quality, but at least others are easier to understand.
Was it the intention of the Authors to mark references 25 and 30 as it is in the manuscript?
Why is most of page 4 empty? The same with page 7.
Reviewer 2 Report
Comments and Suggestions for Authors
The authors report DFT studies of methanol and related species' adsorption and reactivity on a PtSn alloy with the goal of explaining the mechanism and observed experimental differences in CO poisoning behavior between PtSn and Pt for the methanol steam-reforming (MSR) reaction. The results argue convincingly that the favored mechanism on the studied PtSn alloy involves H2O decomposition at the rate-determining step, which in turn explains the higher resistance to CO poisoning.
Overall, the work appears to have been completed in keeping with best practices in the field and, to the best of my knowledge as someome versed in DFT applications to heterogeneous catalysis but not specifically to DMFC chemistry, delivers a new insight that should prove useful to the field of DMFC research. I am supportive of its publication in Nanomaterials.
The summary of prior DFT mechanistic studies of methanol oxidation on is thorough but not as pedagogically accessible as it should be. In other words, the summary succinctly summarizes, one sentence at a time, a series of different studies, but makes no effort to synthesize common conclusions or draw out contrasts from these studies to better organize the current state of knowledge. This paragraph should be revised to better synthesize the current state of the art.
I agree with the authors' claim that there is no prior study of the COMPLETE MSR process on PtSn alloys, but there are some closely related works that should be cited: see DOI:10.1021/jp049354t and DOI:10.1016/j.jmgm.2023.108621 for examples.
The computational methods section is clear and follows best practices for mechanistic studies of heterogeneous catalysis on metal alloys.
The resolution of embedded text is too poor in certain figures, especially Figure 3; especially the subscript on E_a is barely parsable. Similarly, embedded text in Figure 5 (perhaps the most important figure in the paper) is very difficult to read. Although not a requirement, I believe the mechanisms would be easier to parse if they were plotted separately; there is a cluster of states roughly 2/3 of the way across the fictitious reaction coordinate (really multiple coordinates superimposed on one another, hence the clustering) that is nearly impossible to read because everything is squished so close together.
Minor formatting comments:
At line 163, the intrusion of Table 1 results in a false new paragraph that should be merged with the previous paragraph, as it splits the narrative mid-sentence.
Line 321, "precious" should be "previous" (though the former is acknowledged and may still be true :)
Round 2
Reviewer 1 Report
Comments and Suggestions for Authors
In my opinion the manuscript can be published in the present form.
Author Response
Thanks again for the careful reviewing of this manuscript.